# Focal Lesions of the Liver and Radiomics: What Do We Know?

**DOI:** 10.3390/diagnostics13152591

**Published:** 2023-08-03

**Authors:** Matilde Anichini, Antonio Galluzzo, Ginevra Danti, Giulia Grazzini, Silvia Pradella, Francesca Treballi, Eleonora Bicci

**Affiliations:** Department of Radiology, Careggi University Hospital, Largo Brambilla 3, 50134 Florence, Italy; matilde.anichini@unifi.it (M.A.); antonio.galluzzo@unifi.it (A.G.); grazzini.giulia@gmail.com (G.G.); pradella3@yahoo.it (S.P.); francescatreballi@gmail.com (F.T.); eleonora.bicci92@gmail.com (E.B.)

**Keywords:** liver, contrast-enhanced computed tomography (CT), contrast-enhanced magnetic resonance (MRI), radiomics, texture features

## Abstract

Despite differences in pathological analysis, focal liver lesions are not always distinguishable in contrast-enhanced magnetic resonance imaging (MRI), contrast-enhanced computed tomography (CT), and positron emission tomography (PET). This issue can cause problems of differential diagnosis, treatment, and follow-up, especially in patients affected by HBV/HCV chronic liver disease or fatty liver disease. Radiomics is an innovative imaging approach that extracts and analyzes non-visible quantitative imaging features, supporting the radiologist in the most challenging differential diagnosis when the best-known methods are not conclusive. The purpose of this review is to evaluate the most significant CT and MRI texture features, which can discriminate between the main benign and malignant focal liver lesions and can be helpful to predict the response to pharmacological or surgical therapy and the patient’s prognosis.

## 1. Are Focal Liver Lesions Always Distinguishable?

Focal liver lesions, such as hepatocellular carcinoma (HCC), hepatocellular adenoma (HCA), liver metastasis (LM), hemangiomas (HH), or focal nodular hyperplasia (FNH), require differential diagnosis, especially in the absence of clinical and laboratory data. In 80% of cases, HCC arises in patients with cirrhosis with previous hepatitis B and C infection or alcoholics and it is the most widespread primary liver cancer [1,2]. In cirrhotic liver, HCC shows hyperenhancement of the arterial phase, followed by a delayed portal or venous phase washout on computer tomography (CT) or magnetic resonance imaging (MRI) with multiphasic contrast [3]. In non-cirrhotic liver cases, differential diagnosis with hypervascular lesions, such as HCA and FNH, is challenging [4,5], especially when they lack typical imaging features, such as a central scar, suggestive for FNH (reported in about 50% of FNHs larger than 3 cm) [6]. LM are 10–40 times more common than HCC [7] and typically in an early-phase contrast-enhanced CT (CECT), an annular enhancement of the focal lesion is observed; then, a central avascular area in the portal phase and delayed enhancement in the central area in the equilibrium phase can be observed. In highly vascularized tumours, instead, LM shows early enhancement in the arterial phase, late homogeneous enhancement in the portal or equilibrium phase, and washout in the equilibrium phase. MRI and its combination with hepatospecific contrast agents and diffusion-weighted imaging (DWI) techniques are the most sensitive and specific techniques to distinguish liver metastasis from other focal liver lesions, but differential diagnosis is not always straightforward, given certain similarities in their radiological behaviour [3,8,9]. FNH is the second most common benign liver tumour after liver cysts and before adenoma [3,10]. It occurs more commonly in women using oral contraceptive in the third–fourth decades of life and presents as a noncapsular mass with a central stellate area of fibrosis and nodular hyperplasia of the liver parenchyma associated with a congenital vascular malformation [11]. MRI has higher sensitivity than CT for the diagnosis of FNH, but in some cases, it may present with atypical features that can generate confusion with HCC, such as a strong hyperintensity on T2-weighted imaging, a pseudocapsule, washout or absence of a central scar in small FNH, and the presence of haemorrhage, calcification, or necrosis [12,13]. HCA is characterized by proliferation of mature hepatocytes and can be complicated by hemorrhage and malignant transformation into HCC. HCA is generally a homogeneous lesion with mild prolonged enhancement and smooth margins. It can sometimes be heterogeneous in T1- or T2-weighted stages, as in cases of hemorrhage or necrosis. It is usually hypointense in a hepatocytic phase on MRI with a hepatospecific contrast agent. These findings do not facilitate differentiation from HCC [13]. They are currently divided into four independent genetic and pathological subtypes: inflammatory HCA, HCA mutated in hepatocyte nuclear factor 1 alpha, HCA mutated in b-catenin, and unclassified HCA. Inflammatory HCA accounts for 50% of all HCAs and is mostly found in women; it is related to the use of oral contraceptives, obesity, diabetes hepatic steatosis, and glycogenosis. [13,14] (Figure 1 and Figure 2).

According to what has been described so far, the aim of this study was to develop a review about CT and MRI features of focal liver lesions (FLL) in order to distinguish malignant from benign lesions earlier and find CT and MRI radiomic features that could have a role in terms of the prognosis and response to medical or surgical therapy treatments.

## 2. Radiomics and Feature Classification

CT texture analysis (CTTA) is an area of “radiomics” that allows the objective assessment of a lesion and organ heterogeneity beyond what is possible with subjective visual interpretation and may reflect information about the tissue microenvironment; CTTA has demonstrated promise in lesion characterization; in particular it is able to differentiate benign from malignant or more biologically aggressive lesions. There are many texture analysis methods, such as statistical-, model-, or transformed-based-methods; statistical-based techniques are generally applied to describe the relationship of grey-level values in an image [15]. Three orders of parameters are described in a statistical-based texture analysis. The first-order statistics are related to gray-level frequency distribution within the region of interest (ROI), which can be obtained from the histogram of the pixel intensities [16]. The first-order parameters are calculated from the original values and do not describe the relationship between the pixels. They include the mean, medium, and maximum intensity, the standard deviation (SD), the skewness (the asymmetry of the histogram), the kurtosis (flatness of the histogram), or the MPP (mean of the positive pixel). The second-order parameters related to the grey-level co-occurrence matrix (GLCM) are entropy, energy angular second moment (ASM), and homogeneity or dissimilarity. The second-order features related to the GLCM count the number of pixel transitions between the two pixel values [17]. Other important second order parameters derive from the grey-level-run-length matrix (GLRLM), which provides information about the number of equal and consecutive grey levels in each course. These are short-run-emphasis (SRE), long-run-emphasis (LRE), grey-level non-uniformity (GLNU), run-length-non-uniformity (RLNU), and run-percentage (RP). Another second order feature is the grey-level size zone matrix (GLSZM), which includes low grey-level zone emphasis (LGZE) and more. Higher-order features are obtained by applying filters (model-based features) or mathematical transformations (transform-based features) to images and provide information on more than two pixels or voxels. The neighbourhood grey-tone difference matrix (NGLDM) is a higher order parameter that corresponds to the difference of the grey level between one voxel and its 26 neighbours in three dimensions [18]. The NGLDM includes features such as coarseness (measurement of edge density), business (spatial rate of the grey-level change), or contrast (number of local variations in an image). Another higher order feature that compares the differences between multiple pixel/voxel is the neighbourhood grey-tone difference matrix (NGTDM). There are a wide variety of imaging filtration methods. A Laplacian or Gaussian bandpass filter is a commonly used advanced image filtration method that alters the image pixel intensity patterns and allows for the extraction of specific structures corresponding to the width of the filter. Lower filter values correspond to fine texture features, while higher filter values emphasise medium or coarse texture features [19]. Model-based approaches represent texture using sophisticated mathematical models, such as fractal analysis. Fractal analysis is a form of pattern or geometric recognition. The fractal dimension is a measure of the irregularity or roughness of a surface [20,21]. Transform-based methods (Fourier, Gabor, and wavelet transforms) analyse the texture in a frequency or the scale space [19]. Fourier transform analyses the frequency content without spatial localization but is used as frequently. Gabor transform is a windowed-Fourier transform derived by the introduction of a Gaussian function, which then allows for frequency and spatial localization, but this method is limited by its single filter resolution. This problem is overcome by wavelet transform, which uses multiple channels tuned to different frequencies [21]. In more recent years, texture analysis has also been applied to MRI; texture features can be derived from the grey-level histogram, GLCM, or run-length-matrix (RLM), as we saw for CTTA. Other texture features derive from the absolute gradient (gradient mean, variance, skewness, kurtosis, and non-zeros), autoregressive model, or wavelet transform [22] (Table 1, Table 2 and Table 3).

## 3. Malignant CT Features

### 3.1. FLL Feature Characteristic

In 2007, Mougiakakou et al. proposed five different architectures obtained by combining texture features with an ensemble of classifiers (EC) in order to optimise the performance of computer-aided diagnosis (CAD) systems. The best performing architecture was able to classify normal liver tissue, liver cysts, HH, and HCC from non-enhanced CT images with an average classification accuracy of 84.96% [23]. Raman et al. obtained two high-sensitivity and -specificity prediction radiomic models both based on first-order features (mean and SD). The first was able to differentiate a liver without any lesions from a liver with FNH, HCA, or HCC; these two characteristics alone highlight the shared hypervascularity of the lesions. The other model defines whether or not an HCC is present in the liver [24].

### 3.2. FLL CT Radiomic Features and Prognosis

Radiomic models are able to predict patient outcomes, as showed by Lubner et al., who correlated CT texture features of pre-treatment hepatic metastatic disease with pathologic features and clinical outcomes. They analysed 77 non-treated patients with colon–liver metastasis (CLM) and found an inversely related association between the tumour grade and entropy, SD, MPP. They did not find any significant association between the baseline serum carcinoembryonic antigen (CEA) and texture features. The skewness and kurtosis showed association with KRAS, although it was only available in about half the patients. Their data express how greater homogeneity causes a greater tumour grade and worse survival, different to other studies wherein there was a correlation with the tumour heterogeneity, hypoxia, and neoangiogenesis, and, thus, to a decrease in MPP and entropy [25,26,27,28,29]. Radiomics may be also able to identify patients at high risk for the development of colorectal–liver metastasis (CRLM) at the first diagnosis stage. In a recent study, a radiomic model based on 101 features was developed and two feature were the most relevant: the median and small-dependence low grey-level emphasis (SDLGE). This model, if combined with clinical features, outshines the clinical model itself [30]. A combination of the clinical and radiomic features can help to predict the recurrence of HCC in preoperative CT better than clinical variables alone [31]. An example was given in a recent study by Gu-Wei Ji et al. where they aimed to establish the recurrence risk in HCC models based on the radiomics features meeting the Milan criteria in patients undergoing resection. They built a preoperative combined model including a radiomics signature and clinical radiologic parameters available before surgery, and a postoperative combined model, which included the aforementioned predictors plus pathologic variables. Two clinical models were generated on the basis of the semantic features and parameters available before or after surgery. Both the prognostic performance of the preoperative and of the postoperative radiomic model was superior to the preoperative clinical models. Among thousands of radiomic features identified, an accurate selection the wavelet-based features achieved the highest weights in order to build the radiomic signature; according to the authors, these features may reflect the spatial heterogeneity of a tumour and its periphery [32]. In the study by Oh et al., when only second-order textural features were included, the skewness was the most commonly identified feature predictive of the outcomes. They also reported that the skewness predicted the overall survival (OS) better than microvascular invasion [33]. Defour et al. performed a multivariable analysis of textural features in the portal-venous phase and found that the skewness was strongly associated with the OS [34]. Kim et al. found that a high-order feature analysis performed similarly to a combined clinical model (age, hepatitis C, alcohol use, cirrhosis, tumour capsule, and microvascular invasion) in predicting early recurrence (ER). The authors also demonstrated that the inclusion of 3 mm of peritumoural tissue improved risk prediction over segmenting the tumour alone [35]. Lubner et al. also investigated the role of CTTA in the prediction of the pathology and clinical outcomes in patients with LM from CRC: entropy, MPP, and SD at medium filtration levels were significantly associated with the tumour grade, while the skewness was negatively associated with KRAS mutation. Entropy at coarse filtration levels was associated with survival [29]. Some studies demonstrated an association between the homogeneity/heterogeneity of LM and survival. Ravanelli et al. reported a lower OS and progression-free survival (PFS) in patients with a higher uniformity of the CRLM CT scan [36]. These results agreed with Andersen et al., who described an association between the shorter OS for patients affected by CRLM and the tumour homogeneity on the CT [37] (Figure 3). 

### 3.3. LFF CT Radiomic Features and Response to Therapy

In the response to therapy field, Mulé et al. investigated the role of pre-treatment CTTA in predicting the OS and the time to progression (TTP) in patients affected by advanced HCC treated with sorafenib; portal phase-derived entropy at fine-, medium-, and coarse-texture scales was identified as an independent predictor of OS. Entropy at the portal-phase could be a predictor of survival in patients with advanced HCC treated with sorafenib [38]. Regarding the prediction of the response to therapy and prognosis of LM, Su Joa Ahn et al. analysed a cohort of 235 patients with LM from CRC who underwent CT and cytotoxic chemotherapy using FOLFOX and FOLFIRI; they found that lower skewness in 2D, higher mean attenuation, and narrower SD in 3D were independently associated with the response to chemotherapy [39]. With the support of artificial intelligence, radiomics can generate automated models. Hu et al. aimed to develop a prediction model utilising radiomic features from liver volumes as input data to machine learning models in order to predict patient outcomes in patients with CLM treated with radiotherapy. The most predictive radiomic feature was NGTDM strength [40]. Klassen et al. investigated if a CT radiomics approach could predict the response of individual LM of esophageal cancer in patients treated with chemotherapy (CAPOX); most of the extracted features correlated to heterogeneity or described the tumour intensity and seemed to predict the response to therapy. Another paper focused on non-CRLM in esophageal cancer; the study found that the characteristics of pre-treatment CT related to heterogeneity and the grey-level intensity, such as wavelet grey-level co-occurrence matrix correlation and grey-level distance zone matrix with large dependence emphasis, were predictors of the response to chemotherapy [41]. As for non-CRLM, Martini et al. analysed a small series of patients and observed a number of associations: pancreatic NET had a lower skewness and higher mean Hounsfield (HU) than non-pancreatic ones; entropy in the arterial phase was negatively associated with PFS in pancreatic NET and with OS in non-pancreatic NET; and kurtosis was associated with a lower OS in pancreatic NET, while skewness was associated with a higher one [42] (Figure 4). 

## 4. Benign Radiomics CT Features

As mentioned above, there are some imaging characteristics about focal liver lesions that may overlap. Regarding the distinction between HCA and FNH, Cannella et al. reported a strong difference in the CTTA features between these two groups that may reflect the different histopathology of these lesions. In particular, on arterial phase images, the mean, mpp, and skewness were significantly higher in FNH than in HCA on the unfiltered images; the SD, entropy, and mpp were higher on the filtered analysis. On the portal-phase instead, the mean, mpp, and skewness in FNH were significantly different from HCA on the unfiltered images, while entropy and kurtosis were significantly higher in FNH on the filtered images [43]. Raman et al. investigated the role of CTTA in differentiating a hypervascular liver lesion, including FNH, HCA and HCC. The study showed that only the mean in medium (3 mm and 4 mm) and coarse filters (5 and 6 mm) was statistically different in distinguishing FNH from HCA, only on arterial phase images [24]. As described above, Nie et al. built a radiomic signature using 10 features to distinguish FNH and HCC; the radiomic score showed a statistically significant difference between these two lesions. In another study the same author built a radiomic nomogram, which incorporated the patient’s gender, age, enhancement pattern, and radiomics score (made by a radiomic signature from seven features). The radiomics nomogram was better able to distinguish HCC from HCA in a non-cirrhotic liver than the radiomic signature alone [44]. Hu et al. built a radiomic index on an unenhanced CT using two features (wavelet-LLL first-order median and wavelet-LHL-GLSZM-zone entropy) that showed great performance in differentiating HH from HCC; lower GLSZM zone entropy and higher median values of the voxel intensity values indicate more uniform pixels in the region of interest, and these results might be highly consistent with the pathological differences between HH and HCC, in which HH consists of a vascular malformation and HCC contains mainly cytological atypia [45]. Song et al. investigated the ability of CTTA to distinguish different hypervascular hepatic focal lesions by dividing the benign lesions (HH, HA, FNH) from the malignant ones (HCC, LM). They found that seven texture features (max intensity, range, kurtosis, quantile 95, min size, sum variance, and inverse difference moment) showed significant differences between the benign and malignant groups [46]. Xue et al. also built a radiomic-based model to differentiate intrahepatic-cholangiocarcinoma (I-CHC) from an inflammatory mass with hepatolithiasis by selecting two features from the arterial phase: (1) GLCM-correlation and (2) grey-level zone length matrix (GLZLM) and three features from the portal phase (SHAPE compacity, NGLDM contrast, GLZLM); both features from the arterial and venous phases could differentiate these two types of lesions. The diagnostic accuracy improved with the clinical data [47]. Zhao et al. selected twenty-seven radiomic features with specific associated filters to distinguish six categories of FLL (HCA, HH, FNH, cysts, HCC, metastasis); the classifiers had good diagnostic performance, with the area under curve (AUC) values greater than 0.900 in the training and validation groups [48]. A Pyogenic hepatic abscess may mimic primary or secondary carcinoma of the liver on CECT [49]. Hepatic abscesses usually appear as thick-walled lesions with low attenuation on CT and show increased peripheral rim enhancement on CECT [50]. However, imaging findings are often nonspecific because certain primary or secondary carcinomas of the liver may develop central necrosis, which may mimic the appearance of hepatic abscesses [51]. Suo et al. demonstrated that there were significant differences in the entropy and uniformity among hepatic abscesses, malignant mimickers, and simple cysts. The abscess had a significantly higher entropy and lower uniformity compared with malignant mimickers, since the hepatic abscess was more radiologically heterogeneous than the malignant mimicker [49] (Figure 5).

## 5. Malignant MRI Features

### 5.1. FLL Feature Characteristic

Some radiomic features can provide information about the genomic, proteomic and transcriptomic characteristics. Granata et al. were the first to achieve, through thought radiomics, a correlation between RAS mutation and liver metastasis in a multivariate analysis: contrast, dissimilarity, and entropy were the most significant features [52]. Hectors et al. not only investigated the correlation between MRI histogram features and immunohistochemical markers of HCC, but also evaluated how potentially quantitative radiomics analysis can non-invasively predict immuno-oncological features and HCC. They found that an ADC map variance and enhancement ratios in portal and late venous phases were significantly correlated with PD-L1 checkpoint inhibitor expression and several texture features associated with CD 68. ADC min was associated with more or less aggressive molecular subtypes, and late arterial phase textures were related to PD-1 and CTLA4 immunotherapy targets, while sum entropy was significantly associated with the risk of recurrence [53,54]. These models, if applied to clinical practice, can change patient management and therapy. Li et al. developed the first radiomic model SPAIR T2W-MRI-based concerning 162 patients in a retrospective analysis with the aim to distinguish liver lesions. The radiomic features selected had three peculiarities, namely, reproducibility, high degree of differentiation, and low redundancy. Several features, in particular, the mean of energy, homogeneity, inverse difference moment, inverse variance, small gradient emphasis, gradient non-homogeneity, large gradient emphasis, and gradient entropy differentiated between LM and HCC. Inverse variance, contrast, small gradient emphasis, gradient non-homogeneity, LRE, and long run low grey-level emphasis (LRLGLE) are also able to distinguish HH and HCC; another contribute to this last differentiation may be given by small gradient emphasis and gradient non-homogeneity. SRE and short run high grey-level emphasis (SRHGE) in different directions can differentiate HH and LM [55]. By exploiting the radiomic features obtained from the pre-contrast MRI sequences T2w, out-phases T1W and DWI, Wu et al. attempted to distinguish HCC and HH. The shape features showed the worst results because the shape and volume change during lesion evolution. T2 and the tut-phases sequences showed the best results because of the different fat signals between HCC and HH. The model’s most significant features concerned randomness (entropy, GLCM—difference entropy), flatness (kurtosis), asymmetry (skewness, GLCM—cluster shade), variation (standard deviation, GLRLM—run variance), groupings of voxels with similar grey levels (GLCM—cluster tendency), and heterogeneity (GLSZM—zone entropy) in the ROIs [56]. Regarding the application of contrast medium (CM), Stocker et al. aimed to find diagnostic MRI radiomic features on MRI in order to differentiate between HCC and benign hepatocellular tumours in a non-cirrhotic liver. In particular, five features were significant in the arterial phase (skewness, LGRE, SRLGE, SRHGE, and LRLGE) and two in the portal-venous phase images (LGRE and SRLGE). Only one was significative in the native T1w images (skewness). In particular, the best results were obtained from the arterial phase images because of the different vascular supply of the HCC, as well as the formation of leaky vessels in HCCs. Moreover, HCC and HA in venous and later phases show a similar drainage pattern and the absence of functional hepatocytes [57]. Through the minimum redundancy maximum relevance (mRMR) and the elastic network algorithm method, it is possible to select different significant features from DWI and CE-MRI in order to distinguish HCC, mass-type cholangiocarcinoma (MCC), and CHC, as demonstrated by a recent study, which found the cluster prominence, uniformity, and GLCM energy to be the relevant features. These features were common for three nomograms, which differed for the sequences included [58]. Zhang et al. aimed to discriminate between HCC and non-HCC in LM using radiomics features extracted from an MRI. Different models were built, and some features were common, such as minimum, skewness (first-order feature), inverse different moment normalised (second-order feature), and flatness (shape feature). Furthermore, a wavelet transform was applied to decompose the original image in order to obtain wavelet-based features; different first-order (mean, 10th percentile, kurtosis, robust mean absolute deviation), GLCM (Idn, Imc1, MCC, and dependence variance) NGTDM (strength), and GLSZM (large area low grey-level emphasis) features became important in different models. The model based on T2W and contrast-enhanced T1W images achieved the best discrimination performance [59,60]. Xuehu Wang et al. developed MRI-based radiomic models involving both low-order and high-order features to distinguish combined hepatocellular cholangiocarcinoma (cHCC-CC), HCC, and CHC. A model based on higher-order features significantly improved the diagnostic capability compared to a low-order features model because of its susceptibility to noise. The most influential feature was LRLGLE, which received the highest weight in the lasso regression [61]. Yang et al. developed and validated a radiomics nomogram for preoperative prediction of microvascular invasion (MVI) in HCC involving 208 patients. Their combined model showed that 10 radiomic features were correlated with MVI, in particular, the sphericity (MVI correlates with capsular invasion and, consequently, with an irregularity of the margins), root mean square (different tumour areas correspond to different grey levels depending on whether or not MVI is present, and, consequently, to a different root mean square), and median of the intensity histogram (this feature is reduced if MVI is present, as seen in other studies) [62,63]. A CAD system was used to automatically classify the focal lesions of the liver segmented automatically by algorithms on the T2W images in the study by Gatos et al. Contrast features, such as the inverse different moment, sum variance, and LRE, were found to be the most accurate features of the classification, none being correlated with lesion shape or morphology [64]. In 2018, the American College of Radiology developed standardised criteria to define the probability of malignancy of a liver lesion: The Liver Imaging Reporting and Data System (LI-RADS). Among the various categories, LI-RADS M indicates a malignant lesion where there is no absolute certainty of it being an HCC. Such lesions are categorised as malignant by MRI and include not only HCC, but also ICHC, cHCC-CC, and metastases [59,65]. A radiomic nomogram model based on eight texture features (vertical run-length nonuniformity, difference variance, sum of squares, wavelet energy, LL scale1, sum entropy, sum variance, sigma) extracted from T1-weighted, T2W, and apparent diffusion coefficient (ADC) images improved the diagnostic accuracy of LI-RADS in benign and malignant liver lesions differentiation [66]. As previously explained, radiomic models may potentially predict the occurrence of LM: Shou et al. aimed to develop a radiomics signature based on primary rectal cancer for the preoperative prediction of synchronous LM. The radiomic score and tumour stage on MRI (mT-stage) were identified as independent predictors of synchronous LM in patients with rectal cancer. A nomogram incorporating these two factors in order to predict synchronous LM was constructed and achieved a maximum sensitivity of 73.11%. This value is lower than the diagnosis obtained by CE-MRI, but the potential savings in time and money were emphasised [67]. Liu et al. built a radiomic signature based on five radiomic features obtained from preoperative T2 images (coarseness, cluster shade, high grey-level zone emphasis, median, and dependence variance) able to predict CLM. This model was less accurate than a nomogram obtained by combining the radiomic signature with CEA and CA-19.9 [68]. Few studies analysed the repeatability of the radiomic feature in MRI [69], but Carbonelli et al. were the first to attempt to assess radiomic feature reproducibility on multiple MRI sequences in a normal liver and HCC. Although there was an acceptable repeatability using the same MRI system and across readers, a decrease in the inter-platform reproducibility of first- and second-order radiomic features between the MR sequences, with a less pronounced decrease on the T1WI sequences was registered. The decline was supposedly related to different acquisition parameters, reconstruction, and field strength variation between the MRI systems from the same or different vendors [70]. Several studies have been listed that have proposed an automatic classification of liver lesions, but Jansen et al. proposed a method able to differentiate between five lesion classes, namely, HCA, cysts, HH, HCCs, and metastases, by exploiting features derived from delayed CE-MR images with an extracellular contrast agent, as well as features from T2-weighted images. Importantly, due to the low sample size, FNH was not included. The risk factors for adenoma, HCC, and metastasis were also taken into account as features. They tested how the addition of these features from the delayed CE-MR images and the risk factors from the T2-weighted MR image features improved FLL classification. In particular, for both grey-level histograms (mean, 10th perc., 90th perc., SD, skewness) and texture features (sum of squares variance, sum of average sum variance, IMC1, correlation, sum entropy, difference variance, contrast), the addition of CM expanded the number of useful radiomic features. They were able to differentiate between malignant and benign lesions with a sensitivity of 0.92 for benign lesions and 0.86 for malignant lesions. The specificity was 0.91 and 0.88 for benign and malignant, respectively [71] (Figure 4).

### 5.2. LFF MRI Radiomic Features OS and Response to Therapy

Several studies have evaluated the role of radiomics in the prediction of HCC prognosis using MRI features. In a recent prospective study, the efficiency of CE-MRI-based radiomics features for the prediction of OS in HCC patients after surgical resection was evaluated. The radiomic signature alone distinguished high-risk from lower-risk survivors with HCC and when combined with the clinical features in a combined model, the predictive ability was improved [72,73]. In another study, Zhang et al. investigated the value of texture analysis and conventional MRI features for predicting the ER of single HCC after hepatectomy; a total of 100 HCC patients were divided into two groups (A and B based on tumour diameter > or <3 cm) and then classified into two subgroups with ER or non-ER. In a multivariate logistic regression analysis, uniformity and entropy based on arterial phase images and an irregular margin in group A, and skewness and entropy based on arterial phase images and arterial peritumoural enhancement in group B were independent predictors for ER. Entropy displayed a higher predictive power for ER. I-CHC is an aggressive primary hepatic cancer arising from the bile duct epithelium; surgical resection is currently the only curative treatment [74,75]. A recent single-centre retrospective study reported that the radiomics signature on preoperative arterial-phase contrast-enhanced MR images can be used to predict early recurrence of I-CHC after partial hepatectomy with an AUC of 0.82 and 0.77 in the training and validation cohort, respectively [76]. Song et al. showed that features extracted from pre-treatment portal venous phase MRI images could be useful to build a combined model together with clinical information that is able to evaluate the recurrence-free-survival (RFS) of patients with HCC who undergo c-TACE [77]. Trebeschi et al. reported heterogeneity-related radiomics parameters as predictors of the response to immunotherapy in LM of melanoma and non-small-cell lung carcinoma (NSCLC); he found that lesions responding to anti-PD1 antibodies in a pre-treatment CT presented higher levels of irregular patterns (wavelet, HLH-GLSZM-ZoneEntropy) with more compact, spherical profiles (surface–volume ratio). This result suggests that morphological heterogeneity does not necessarily correspond to a genetic heterogeneity [78].

## 6. Benign MRI Features

Cannella et al. assessed the performance of a texture analysis on CE-MRI for differentiation of HCA from FNH. The skewness on T2-weighted imaging and entropy on hepatobiliary phase imaging (HBP) were significantly higher in FNHs than in HCAs. Furthermore, the skewness on arterial phase imaging and the skewness on HBP imaging were significantly higher in HCAs than in FNHs. A value of skewness greater than −0.06 had a sensitivity of 72.5% and a specificity of 90.6% for the diagnosis of HCA and was the most relevant for diagnosis [79]. Zhao et al. developed a radiomics model based on triple-phase CE-MRI images to differentiate between fat poor angiomiolipoma (fp-AML) and HCC in a noncirrhotic liver. The radiomics features that contributed most to the diagnosis of fp-AML were root mean squared, mean, and 90th percentile [80]. In addition, Ding et al. [81] aimed to develop and validate a radiomic model for differentiating HCC from FNH in non-cirrhotic livers using CE-MRI. In his study, the author built a radiomic model based on eight features to obtain a radiomic score; a clinical model (including sex, HbSAg, and enhancement pattern) and a combined model were also established. The patients were randomly allocated to a training and validation set; both these groups benefited from the combined model, which had higher accuracy. In both groups, when comparing the AUCs between the three models, the combined model proved to be significantly better. The difference between the clinical model and radiomics model was not statistically significant (Figure 6).

## 7. Conclusions

Taking into account the difficulties in distinguishing between benign and malignant liver lesions with similar imaging characteristics, analysing some quantitative features not visible to the naked eye could represent a breakthrough. The studies reviewed thus far have demonstrated the superiority of a combined clinical and radiomic model in predicting the classification, prognosis, and response to therapy of FLL compared to the clinical model alone. Unfortunately, radiomics nowadays lacks standardised models, although it has potential applications in imaging.

## Figures and Tables

**Figure 1 diagnostics-13-02591-f001:**
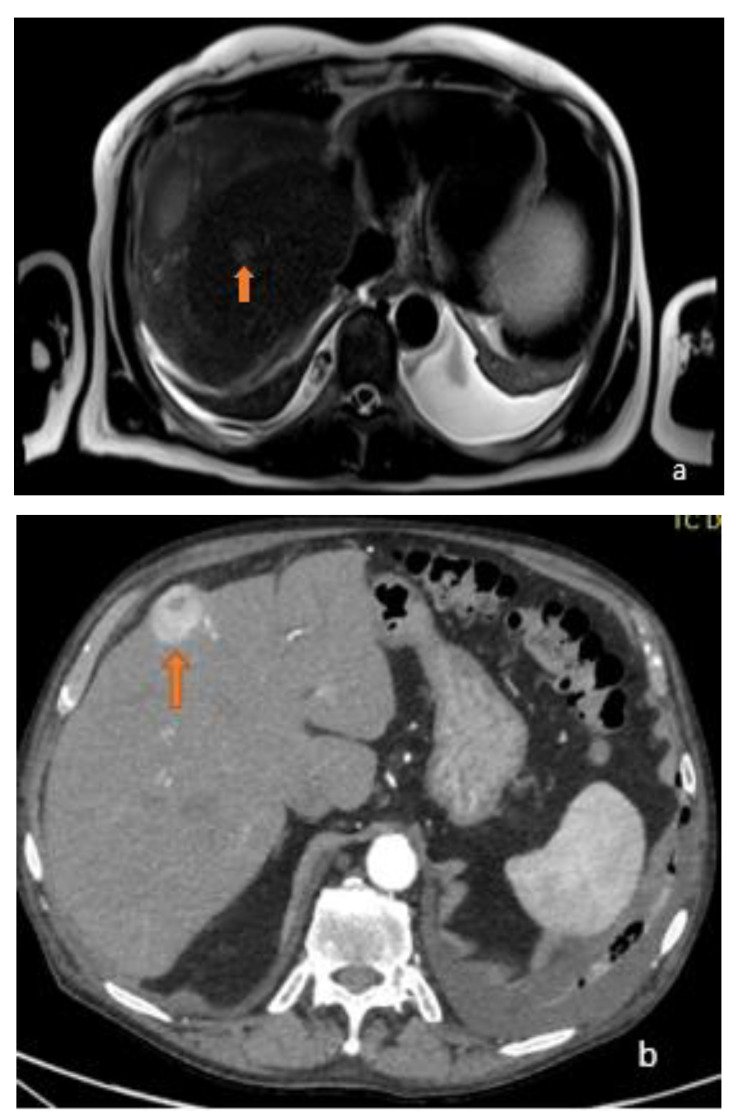
(**a**) Axial T2-weighted MRI image and (**b**) axial arterial-phase CT image show two different focal lesions (orange arrows), respectively, hyperintense in the T2-weighted sequences and endowed with marked wash-in in the contrastographic arterial phase CT. Both lesions proved to be HCC on biopsy examination.

**Figure 2 diagnostics-13-02591-f002:**
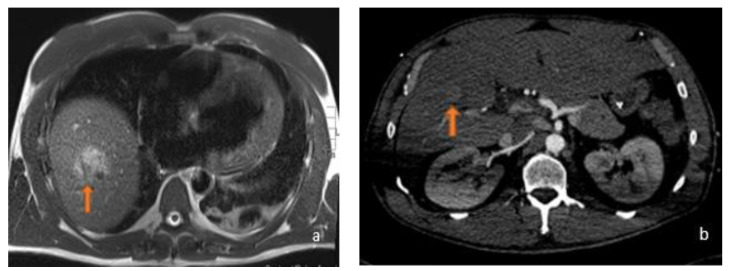
(**a**) Axial T2-weighted MRI and (**b**) axial arterial-phase CT image belonging to the same patient: this was a 45-year-old male who was using anabolic steroids. In relation to the patient’s history, which excluded a history of potus or hepatotropic virus infection, the supposed diagnosis was inflammatory adenoma. As can be observed, the lesions (orange arrows) appeared, for HCC, hyperintense in T2-weighted sequences with a strong contrastographic enhancement in the arterial phase CT.

**Figure 3 diagnostics-13-02591-f003:**
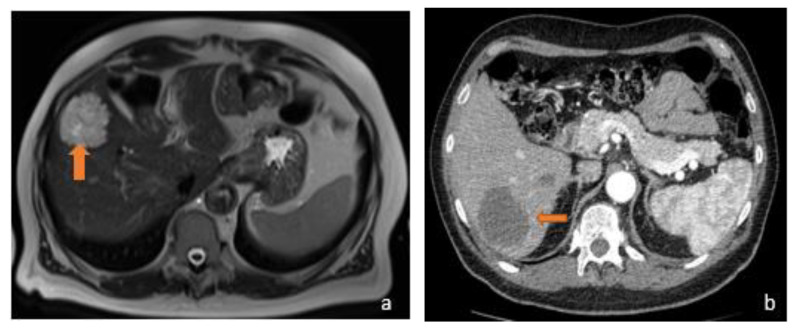
(**a**) Axial T2-weighted MRI and (**b**) axial arterial-phase CT image belong to the same patient, suffering from metastatic mucinous colorectal carcinoma. MRI examination shows a hyperintense focality in the T2-weighted sequences and a hypodense lesion in the arterial phase on CT investigation (orange arrows). The reason why the lesions exhibit this signal and contrastographic behaviour stems from their mucinous content.

**Figure 4 diagnostics-13-02591-f004:**
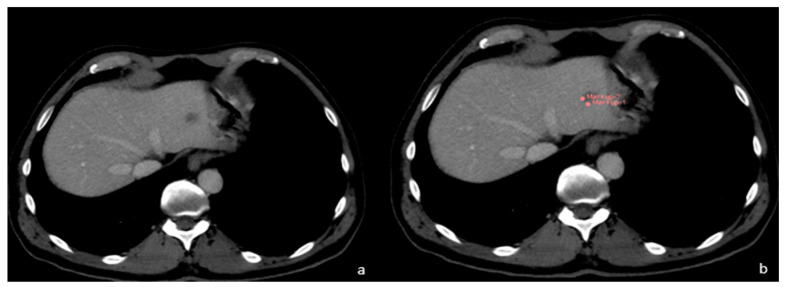
(**a**) FLL of unknown origin. (**b**) First phase of FLL segmentation in CT.

**Figure 5 diagnostics-13-02591-f005:**
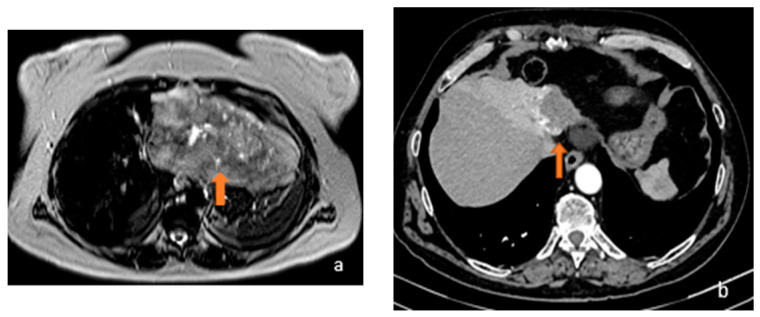
(**a**) Axial T2-weighted MRI and (**b**) axial arterial-phase CT image show, respectively, a gross inhomogeneously hyperintense focality on T2-weighted sequences and a hypodense lesion in the CT arterial phase characterised by centripetal contrastographic enhancement (orange arrows). Both lesions, given the signal on MRI and the contrastographic behaviour on CT, were considered to be angiomas.

**Figure 6 diagnostics-13-02591-f006:**
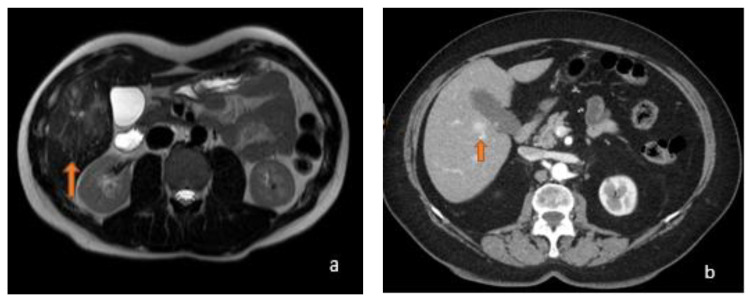
(**a**) Axial T2-weighted MRI shows an image (orange arrow) with a hyperintense stellate scar in T2-weighted sequences, a typical appearance of FNH, with a central fibrotic area that appears hyperintense in T2 and hypointense in T1. (**b**) Axial arterial-phase CT shows a focality in the V hepatic segment at pericholecystic site (orange arrow) characterized by rapid filling in arterial phase contrastographic on CT; FNH may show this contrastographic behaviour.

**Table 1 diagnostics-13-02591-t001:** Description of Radiomic Features.

Texture Features (Statistical-Based Methods)	Level	Description	Examples
Intensity of the pixel/voxel histogram	First	Grey-level frequency distribution from the pixel intensity histogram in a region of interest (ROI). First-order parameters are obtained from original values; they do not describe the relationship between pixels.	- Mean, medium, maximum intensity. - Median - Standard deviation - Skewness (asymmetry of the histogram) - Kurtosis (peakedness/flatness of histogram) - Mean of positive pixels - First-order entropy
GLCM (Grey-level co-occurrence matrix)	Second	GLCM parameters count the number of pixel transitions between two pixel values.	- Homogeneity - Energy - Angular second moment - Contrast - Correlation - Second-order entropy - Dissimilarity
GLRLM (Grey-level run-length matrix)	Second	Number of equal and consecutive grey levels in each course.	- Short/long-run emphasis - Grey-level uniformity for run - Run length non-uniformity - Run percentage
NGLDM (neighbourhood grey-level different matrix)	Higher (third or more)	Grey-level difference between one voxel and its 26 neighbours in three dimensions.	- Business - Contrast - Coarseness
Advanced metrics	Higher order	Comparing differences and relationships between multiple pixels/voxels.	- Autoregressive model - Haar wavelet (wavelet energy) - Neighbourhood grey-tone difference matrix (NGTDM)

**Table 2 diagnostics-13-02591-t002:** Examples of CT features.

	Examples	Description
Texture features (model-based methods)	- Fractal analysis	Fractal dimension measures the irregularity or roughness of a surface.
Texture features (transformed-based methods)	- Fourier transform - Gabor transform - Wavelet transform	They analyse texture in a frequency or the scale space.

**Table 3 diagnostics-13-02591-t003:** Examples of MRI features.

Texture Features	Examples
Grey-level histogram	- Mean - Variance - Percentiles (1%, 10%, 50%, 90%, 99%)
Co-occurrence matrix	- Sum entropy - Sum of squares - Inverse difference moment - Difference variance
Run-length matrix	- Run-length non-uniformity - Grey-level non-uniformity - Long/short run emphasis
Autoregressive model	- Teta - Sigma
Wavelet transform	- Energies of wavelet: transform coefficients in subbands LL, LH, HL, HH.

## Data Availability

Not applicable.

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
