# Peer review of "Focal Lesions of the Liver and Radiomics: What Do We Know?"

_diagnostics, 2023, doi:10.3390/diagnostics13152591_

Round 1
Reviewer 1 Report
Thank you very much for the opportunity to review this manuscript.
The authors present a narrative review of radiomics for focal liver lesion analysis on CT and MRI. I suggest below some improvements that could be made:
Abstract: "Despite having a specific behavior..." suggests that all focal liver lesions have the same pattern, which is not correct. I advise rephrasing this sentence to more clearly demonstrate that focal liver lesions can appear similarly on CT and MRI even when they are clearly different on pathologic analysis.
Subheading 1:
While the imaging appearance of focal liver lesions is certainly the main factor considered in the differential diagnosis, other data points such as presentation, age, sex, and history/prior imaging are taken into account by the evaluating radiologist. Thus, epidemiologic features could be included in the description of focal liver lesions provided to better contextualize readers. Notable examples include oral contraceptive use and hepatic adenomas and a greater incidence of FNH in females, while HCC tends to be more common in males.
Figures:
Figure 1 - The presented MRI is severely degraded by motion. Moreover, the enhancement is difficult to appreciate. I would suggest using alternative images or improving the windowing on this image.
Figures 3 and 5 - Check the positioning of the arrow in (a) to more clearly denote the lesion.
Figure 7 is not intelligible on its own, I would suggest making a schematic to improve reader understanding and providing more context in the figure legend.
Tables:
When using bullet points, make them left-aligned to improve readability.
Subheading 2:
When mentioning reference 23, please be more descriptive since most reader will not be familiar with "CAD 4" as mentioned.
Mean and standard deviation are statistical summary measures that can be applied to the variable of interest, please clarify which variable is referred to when citing Raman et al.
On page 7, it is implied that homogeneity caused worse tumor grade and survival, however, causality cannot be inferred from observational studies. Please rectify.
In the sentence "Several radiomic features were identified and Wavelet-based features achieved the highest weights", please provide context to enable readers to understand how this is important to the aforementioned model.
Conclusions:
The conclusion could be more specific on how radiomic analysis represents a breakthrough, for example, by surpassing clinical models alone in predicting lesion classification and treatment response on MRI and CT.
General:
In the manuscript, Computed Tomography is defined as CT and magnetic resonance imaging as MRI, but TC and RM are used several times throughout the paper. Please choose one abbreviation and be consistent with its use. CT and MRI are more common in English.
A short summary could be added to each subheading to summarize the described studies.
There are several grammar and style mistakes that should be revised before publication. I would consider having the manuscript reviewed by a specialized editor or a native speaker to improve the text's readability.
Author Response
Thank you very much for the opportunity to review this manuscript.
The authors present a narrative review of radiomics for focal liver lesion analysis on CT and MRI. I suggest below some improvements that could be made:
Abstract: "Despite having a specific behavior..." suggests that all focal liver lesions have the same pattern, which is not correct. I advise rephrasing this sentence to more clearly demonstrate that focal liver lesions can appear similarly on CT and MRI even when they are clearly different on pathologic analysis.
We thank the reviewer for the suggestion; the sentence has been improved in order to clarify that not all focal liver lesions are well distinguishable on imaging.
Subheading 1:
While the imaging appearance of focal liver lesions is certainly the main factor considered in the differential diagnosis, other data points such as presentation, age, sex, and history/prior imaging are taken into account by the evaluating radiologist. Thus, epidemiologic features could be included in the description of focal liver lesions provided to better contextualize readers. Notable examples include oral contraceptive use and hepatic adenomas and a greater incidence of FNH in females, while HCC tends to be more common in males.
We thank the reviewer for the suggestion; epidemiological data have been added to support the differential diagnosis
Figures:
Figure 1 - The presented MRI is severely degraded by motion. Moreover, the enhancement is difficult to appreciate. I would suggest using alternative images or improving the windowing on this image.
We thank the reviewer for the suggestion; we chose another better-quality image in terms of both motion and contrast.
Figures 3 and 5 - Check the positioning of the arrow in (a) to more clearly denote the lesion.
We thank for the suggestion; we changed the position of the arrows of figure 3a and 5a to clearly denote the lesions.
Figure 7 is not intelligible on its own, I would suggest making a schematic to improve reader understanding and providing more context in the figure legend.
We thank the reviewer for the suggestion, the figure have been removed because it is complex to explain and outside the topic of interest
Tables:
When using bullet points, make them left-aligned to improve readability.
We thank the reviewer for the suggestion, tables have been improved.
Subheading 2:
When mentioning reference 23, please be more descriptive since most reader will not be familiar with "CAD 4" as mentioned.
We thank the reviewer for the suggestion, the sentence has been modified in order to be more comprehensible
Mean and standard deviation are statistical summary measures that can be applied to the variable of interest, please clarify which variable is referred to when citing Raman et al.
We thank the reviewer for the suggestion, the sentence has been modified to clarify that the two models were based both on Mean and SD.
On page 7, it is implied that homogeneity caused worse tumor grade and survival, however, causality cannot be inferred from observational studies. Please rectify.
We thank the reviewer for the suggestion , the purpose of the paragraph "FLL CT radiomic features and prognosis" was to objectively report the studies on the subject. We have tried to eliminate any implicit statements.
In the sentence "Several radiomic features were identified and Wavelet-based features achieved the highest weights", please provide context to enable readers to understand how this is important to the aforementioned model.
We thank the reviewer for the suggestion, the sentence has been modified in order to underline how Wavelet-based features are important to the radiomic model.
Conclusions:
The conclusion could be more specific on how radiomic analysis represents a breakthrough, for example, by surpassing clinical models alone in predicting lesion classification and treatment response on MRI and CT.
We thank the reviewer for the suggestion, the conclusion has been modified in order to specify why radiomics could be a breakthrough.
General:
In the manuscript, Computed Tomography is defined as CT and magnetic resonance imaging as MRI, but TC and RM are used several times throughout the paper. Please choose one abbreviation and be consistent with its use. CT and MRI are more common in English.
We thank the reviewer for the suggestion, the incorrect abbreviations have been changed.
A short summary could be added to each subheading to summarize the described studies.
We thank the reviewer for the suggestion, studies have been summarised in the best way considered
Reviewer 2 Report
Radiomics of focal liver lesion is an evolving area of unmet need. Analysing some quantitative features not visible to the naked eye could represent a breakthrough. The authors have attempted a good review. The following are suggested.
1. TC should be CT
2. RM may be replaced by MRI
3. In the paragraph mentioning “Texture Features (Statistical-based methods)”, the higher order/ advance metrics should be mentioned in more detail.
4. Please explain “GLSZM-zone entropy” before using the ACRONYM.
5. As the data is extensive, it would be better to summarise the details in form of 2 tables. One for CT and one for MRI. The studies correlating the radiomics with RESPONSE and SURVIVAL can be listed individually mentioning the key parameters used, diseases differentiated, and ststistical significance noted in each.
6. Please explain fig 7. What do the values stand for?
Minor english language editing at places
Author Response
Radiomics of focal liver lesion is an evolving area of unmet need. Analysing some quantitative features not visible to the naked eye could represent a breakthrough. The authors have attempted a good review. The following are suggested.
- TC should be CT
We thank the reviewer for the suggestion, the incorrect abbreviations have been changed.
- RM may be replaced by MRI
We thank the reviewer for the suggestion, the incorrect abbreviations have been changed.
- In the paragraph mentioning “Texture Features (Statistical-based methods)”, the higher order/ advance metrics should be mentioned in more detail.
We thank the reviewer for the suggestion; we improved the paragraph describing in more detail the “Higher order/advance metrics” features.
- Please explain “GLSZM-zone entropy” before using the ACRONYM.
We thank the reviewer for the suggestion, the acronym has been added in paragraph 2.
- As the data is extensive, it would be better to summarise the details in form of 2 tables. One for CT and one for MRI. The studies correlating the radiomics with RESPONSE and SURVIVAL can be listed individually mentioning the key parameters used, diseases differentiated, and statistical significance noted in each.
We thank the reviewer for the suggestion, the construction of the suggested table seemed to us, on the contrary, to be an additional complication given the difficulty in explaining the topic, so we considered the paragraphs in question to be exhaustive
- Please explain fig 7. What do the values stand for?
We thank the reviewer for the suggestion, the figure has been removed as it is complex to explain and outside the topic of interest.
Round 2
Reviewer 1 Report
Thank you very much for addressing my comments and suggestions in detail. I believe the changes you implemented led to a significant improvement in the manuscript. I have no other concerns regarding the content.
Minor stylistic and grammar revisions should be addressed before final submission.